# An Adaptive Spatial Target Tracking Method Based on Unscented Kalman Filter

**DOI:** 10.3390/s24186094

**Published:** 2024-09-20

**Authors:** Dandi Rong, Yi Wang

**Affiliations:** Nanjing Research Institute of Electronics Technology, Nanjing 210039, China; wangyi15@nudt.edu.cn

**Keywords:** spatial target tracking, Unscented Kalman filter, adaptive noise factor, cooperation of the space-based infrared satellite and ground-based radar

## Abstract

The spatial target motion model exhibits a high degree of nonlinearity. This leads to the fact that it is easy to diverge when the conventional Kalman filter is used to track the spatial target. The Unscented Kalman filter can be a good solution to this problem. This is because it conveys the statistical properties of the state vector by selecting sampling points to be mapped through the nonlinear model. In practice, however, the measurement noise is often time-varying or unknown. In this case, the filtering accuracy of the Unscented Kalman filter will be reduced. In order to reduce the influence of time-varying measurement noise on the spatial target tracking, while accurately representing the a posteriori mean and covariance of the spatial target state vector, this paper proposes an adaptive noise factor method based on the Unscented Kalman filter to adaptively adjust the covariance matrix of the measurement noise. In this paper, numerical simulations are performed using measurement models from a space-based infrared satellite and a ground-based radar. It is experimentally demonstrated that the adaptive noise factor method can adapt to time-varying measurement noise and thus improve the accuracy of spatial target tracking compared to the Unscented Kalman filter.

## 1. Introduction

As science and technology continue to develop around the world, the strategic importance of the space environment is growing rapidly. In order to understand this space situation in real-time and maintain space security, it is necessary to detect and track spatial targets. Based on the location of the detection sensors, they can be divided into two main categories: ground-based radars and space-based sensors. Ground-based radars provide round-the-clock, all-weather target detection. Space-based sensors are independent of the Earth’s curvature, cover a large area of the Earth and can provide early target information [1]. In this paper, a space-based sensor and a ground-based radar are used together for the detection of a spatial target. 

The space-based sensor used here is an infrared sensor on board a space-based infrared satellite. The infrared sensor is a passive detection system that passively receives the signal emitted, transmitted or scattered by the target. It is a nonlinear system and cannot achieve accurate tracking of the target using conventional Kalman filtering methods. Extended Kalman Filter (EKF), which is more commonly used today, is a first or second order linearized truncation by a Taylor series expansion. This process requires the solving of the Jacobi matrix, which is very complex and tedious, especially in multidimensional systems. In turn, the conventional Kalman filter is used to solve the filtering problem of the linearized systems. This algorithm ignores the higher order terms, while the spatial target motion model has a high degree of nonlinearity. The forced linearization will introduce a greater error, leading to unstable or even divergent filtering results. The Unscented Kalman Filter (UKF), on the other hand, computes the mean and covariance of the posteriori probability density function by unscented transformation(UT) of the deterministic sampling points. This avoids the approximation of the nonlinear function and does not require the solution of the Jacobi matrix. This is more computationally tractable. It is also easy to implement in engineering with high filtering accuracy. Reference [2] used the UKF to perform a UT on a nonlinear system without ignoring higher order terms, which improved the accuracy of the estimation.

However, UKF requires accurate a priori statistical knowledge of the measurement noise. Actually, the statistical properties of measurement noise are unknown or time-varying. This is due to the uncertainty of internal and external influences. This leads to a reduction in the filtering accuracy of UKF in real observations, where UKF has no adaptive processing capability for noise with unknown or time-varying statistical characteristics [3,4,5]. In references [6,7], the observation noise covariance is constant, which cannot truly reflect the dynamic characteristics of the noise. After the observation noise changes, the accuracy of target tracking will decrease.

To address the state filtering problem under the uncertainty of the measurement noise covariance, it is necessary to estimate the measurement noise covariance while updating the state prediction. This is a challenging and significant issue. The algorithm for updating the covariance matrix of measurement noise while updating the estimated state is collectively called the adaptive Unscented Kalman Filter (AUKF) algorithm. Many studies have proposed different AUKF algorithms. Compared with the UKF algorithm, the experimental results of the AUKF algorithm in the environment of time-varying noise measurement are significantly better than the UKF algorithm, and it has better stability and robustness [8,9,10].

Currently, the main method for updating the covariance of measurement noise is the adaptive filtering method based on innovation information [11]. The AUKF algorithm proposed in reference [12] monitors the changes in innovation and residual in the filter and updates the observation noise covariance in real-time using innovation and residual to adjust the gain of the filter and achieve optimal estimation. References [13,14] use measurement residuals to estimate the state vector, which reduces the estimation error. To obtain an improved AUKF algorithm, the output deviation covariance of each measurement is used as the noise covariance in reference [15]. This allows the noise covariance to be updated over time and eliminates the problem of the noise covariance being a constant source of error. In reference [16], the authors estimate and adjust the covariance matrices of the measurement noises according to the covariance matching technique and the innovation and residual sequences. Reference [17] proposes an AUKF algorithm with a sliding window noise estimator.

In this paper, a measurement noise adaptive factor is introduced to estimate the covariance of measurement noise in real time. This factor is updated according to innovation information. The results of Monte Carlo experiments show that the average distance error has been significantly reduced. This algorithm is also able to effectively improve the accuracy of spatial target tracking and maintain better stability and robustness in the situation of large variations in noise characteristics than UKF.

## 2. Spatial Dynamics Model and Measurement Model

This paper uses the Earth-Centered, Earth-Fixed (ECEF) OXYZ coordinate system, which is fixed with respect to the Earth. The origin O of the coordinate system is located at the center of the Earth. The OX axis is in the equatorial plane and points towards the meridian where the Greenwich Observatory is located. The OZ axis is perpendicular to the equatorial plane and coincides with the Earth’s axis of rotation, pointing towards the North Pole. The axes OX, OY and OZ form a right-handed coordinate system [18,19].

Suppose the position r and velocity v of the spatial target in the ECEF coordinate system are as follows: (1)$$\left\{\begin{array}{c} r={\left(\begin{array}{ccc} x & y & z \end{array}\right)}^{T}\\ v={\left(\begin{array}{ccc} vx & vy & vz \end{array}\right)}^{T} \end{array}\right.$$

Then the state vector of the spatial target is $X={\left[\begin{array}{cc} r^{T} & v^{T} \end{array}\right]}^{T}$. The state space model of the spatial target is: (2)$$\dot{X}={\left[\begin{array}{cc} {\dot{r}}^{T} & {\dot{v}}^{T} \end{array}\right]}^{T}+V={\left[\begin{array}{cc} v^{T} & a^{T} \end{array}\right]}^{T}+V$$
where a is the acceleration of the spatial target in the ECEF coordinate system and V is the system model error.

Since the ECEF coordinate system changes with the rotation of the Earth and it is a non-inertial coordinate system, the acceleration of the spatial target in the ECEF coordinate system needs to be corrected by the Coriolis theorem [20]. The acceleration of the spatial target after correction is [21]: (3)$$a=\left[\begin{array}{c} ax\\ ay\\ az \end{array}\right]=\left[\begin{array}{c} -\frac{\mu_{e}}{r^{3}}\left(1+\frac{c_{e}}{r^{2}}\left(1-5{\left(\frac{z}{r}\right)}^{2}\right)\right)\cdot x+\omega_{e}^{2}\cdot x+2\omega_{e}\cdot \dot{y}\\ -\frac{\mu_{e}}{r^{3}}\left(1+\frac{c_{e}}{r^{2}}\left(1-5{\left(\frac{z}{r}\right)}^{2}\right)\right)\cdot y+\omega_{e}^{2}\cdot y+2\omega_{e}\cdot \dot{x}\\ -\frac{\mu_{e}}{r^{3}}\left(1+\frac{c_{e}}{r^{2}}\left(3-5{\left(\frac{z}{r}\right)}^{2}\right)\right)\cdot z \end{array}\right]$$
where $\mu_{e}$ is the gravitational constant of the Earth and is $3.9860064\times 10^{14}\;m^{3}/s^{2}$. r is the distance from the center of the Earth to the spatial target, i.e., the distance from the spatial target to the origin O of the OXYZ coordinate system:(4)$$r=\sqrt{x^{2}+y^{2}+z^{2}}$$

$\omega_{e}$ is the angular velocity in the non-inertial coordinate system. Based on the WGS-84 model, $\omega_{e}=7.292115\times 10^{-5}\mathrm{rad}/s$. $c_{e}=3J_{2}R_{e}^{2}/2$, where $J_{2}$ is the second-order coefficient of zonal harmonics considering the $J_{2}$ perturbation in the ellipsoid model. $J_{2}=1.082626836\times 10^{-3}$. The radius of the Earth’s equator is $R_{e}=6.37814\times 10^{6}\;m$.

According to Equation (3), Equation (2) can be written as: (5)$$\dot{X}=g\left(t,X\right)+V$$
where $g\left(\cdot \right)$ is a nonlinear function. Assuming that the unconsidered effects of perturbation V, including third body gravitational perturbation, solar radiation pressure perturbation, etc., is zero-mean Gaussian white noise, its covariance matrix is Q, i.e., $E\left[VV^{T}\right]=Q$. Since Equation (5) is a nonlinear ordinary differential equation and the derivative and initial value information of the equation is known, the nonlinear continuous-state differential equation can be solved by transforming it into a nonlinear discrete-state equation using the fourth-order Runge–Kutta method, as in Equation (6).
(6)$$X_{k+1}=f\left(X_{k}\right)+V_{k}$$
where $V_{k}$ is the discretized unmodeled systematic error. Under the previous assumptions, $V_{k}$ is a zero-mean white Gaussian process noise sequence. Its covariance matrix is $Q_{k}$, i.e., $E\left[V_{k}{V_{j}}^{T}\right]=Q_{k}\delta_{kj}$. $\delta_{kj}$ is the Kronecker Delta function. 

The relationship between the function $f\left(\cdot \right)$ and the function $g\left(\cdot \right)$ is shown in Equations (7)–(11).
(7)$$f\left(X_{k}\right)=X_{k}+\frac{dt}{6}\left(k_{1}+2k_{2}+2k_{3}+k_{4}\right)$$
where $k_{1}$, $k_{2}$, $k_{3}$ and $k_{4}$ are as follows:(8)$$k_{1}=g\left(X_{k}\right)$$
(9)$$k_{2}=g\left(X_{k}+\frac{dt}{2}\cdot k_{1}\right)=g\left(X_{k}+\frac{dt}{2}\cdot g\left(X_{k}\right)\right)$$
(10)$$k_{3}=g\left(X_{k}+\frac{dt}{2}\cdot k_{2}\right)=g\left(X_{k}+\frac{dt}{2}\cdot g\left(X_{k}+\frac{dt}{2}\cdot g\left(X_{k}\right)\right)\right)$$
(11)$$k_{4}=g\left(X_{k}+dt\cdot k_{3}\right)=g\left(X_{k}+dt\cdot g\left(X_{k}+\frac{dt}{2}\cdot g\left(X_{k}+\frac{dt}{2}g\left(X_{k}\right)\right)\right)\right)$$
where dt is the setting of the sampling interval. 

It should be noted that the Runge–Kutta method is a specific and widely recognized numerical integration method, employed in this paper. Various numerical integration methods are available for solving continuous dynamic equations.

In this paper, a space-based infrared satellite and a ground-based radar are used together for the detection of a spatial target. Space-based infrared satellites can only provide angular information about the spatial target, whereas ground-based radars can provide both range and angular information. Taking into account the angular error and the distance error, the measurement equation is given by Equation (12):(12)$$Z_{k+1}=h\left(X_{k+1}\right)+W_{k+1}={\left[\alpha_{1}\;\beta_{1}\;r_{2}\;\alpha_{2}\;\beta_{2}\right]}^{T}+W_{k+1}$$
where $r_{2}$ is the distance from the ground-based radar to the spatial target. It is assumed that the space-based infrared satellite coordinates are $\left(x_{1},y_{1},z_{1}\right)$ and the ground-based radar coordinates are $\left(x_{2},y_{2},z_{2}\right)$ in the ECEF coordinate system. $\alpha_{i}\left(i=1,2\right)$ and $\beta_{i}\left(i=1,2\right)$ represent the azimuth and pitch angles of the space target relative to each detection sensor, respectively. This is shown in Equation (13). $W_{k+1}$ is the measurement noise. Its covariance matrix is $R_{k}$. In practice, the statistical properties of noise are usually unknown or time-varying. This is due to the uncertainty of internal and external influences on detection. If the measurement noise is assumed to be Gaussian white noise with a constant covariance matrix, the target tracking will have a large error. $Z_{k+1}$ is the observation vector of the spatial target by both detection sensors at the k+1 sampling time.
(13)$$\left\{\begin{array}{c} r_{2}=\sqrt{{\left(x_{2}-x\right)}^{2}+{\left(y_{2}-y\right)}^{2}+{\left(z_{2}-z\right)}^{2}}\\ \alpha_{i}=\arctan\left(\frac{y-y_{i}}{x-x_{i}}\right)\left(i=1,2\right)\\ \beta_{i}=\arctan\left(\frac{z-z_{i}}{\sqrt{{\left(x-x_{i}\right)}^{2}+{\left(y-y_{i}\right)}^{2}}}\right)\left(i=1,2\right) \end{array}\right.$$

## 3. Unscented Kalman Filter 

This paper adopts the Unscented Kalman filter for spatial target tracking. This is because the dynamics model of the spatial target and the measurement model of ground-based radar and space-based infrared satellite have a high degree of nonlinearity.

The state and measurement equations for the Unscented Kalman filter are given in Equation (14):(14)$$\left\{\begin{array}{c} X_{k}=f\left(X_{k-1}\right)+V_{k-1}\\ Z_{k}=h\left(X_{k}\right)+W_{k} \end{array}\right.$$
where $X_{k}$ and $Z_{k}$ are n-dimensional state vectors and l-dimensional observation vectors, respectively, with n=6 and l=5. $V_{k}$ and $W_{k}$ are process noise and measurement noise, respectively. 

Firstly, assume that both the process noise and the measurement noise are independent of each other at different times. Second, assume that the process noise and measurement noise sequences are uncorrelated. Finally, assume that the process noise and the measurement noise are zero-mean white Gaussian process noise. As shown in Equation (15):(15)$$\left\{\begin{array}{l} E\left[V_{k}{W_{j}}^{T}\right]=0\\ E\left[V_{k}{V_{j}}^{T}\right]=Q_{k}\delta_{kj}\\ E\left[W_{k}{W_{j}}^{T}\right]=R_{k}\delta_{kj} \end{array}\right.$$

The Unscented Kalman Filter process is as follows:

Step 1: Set the initial state estimate ${\hat{X}}_{0}$ and initial covariance matrix $P_{0}$
(16)$${\hat{X}}_{0}=E\left(X_{0}\right)$$
(17)$$P_{0}=E\left[\left(X_{0}-{\hat{X}}_{0}\right){\left(X_{0}-{\hat{X}}_{0}\right)}^{T}\right]$$
where $X_{0}$, ${\hat{X}}_{0}$ and $P_{0}$ denote the initial state vector, the estimate of the initial state vector and the covariance of the initial state vector, respectively.

Step 2: Calculate the (2n+1) sampling points $\xi_{k-1}^{i}$ and the corresponding weights
(18)$$\xi_{k-1}^{i}=\left\{\begin{array}{c} \begin{array}{cc} {\hat{X}}_{k-1} & i=0 \end{array}\\ \begin{array}{cc} {\hat{X}}_{k-1}+{\left(\sqrt{\left(n+\lambda \right)P_{k-1}}\right)}_{i} & i=1,\ldots ,n \end{array}\\ \begin{array}{cc} {\hat{X}}_{k-1}-{\left(\sqrt{\left(n+\lambda \right)P_{k-1}}\right)}_{i-n} & i=n+1,\ldots ,2n+1 \end{array} \end{array}\right.$$
where ${\left(\sqrt{\left(n+\lambda \right)P_{k-1}}\right)}_{i}$ is the i-th column of the mean square matrix of matrix $\left(n+\lambda \right)P_{k-1}$. λ is a secondary scale parameter. $\lambda =\alpha^{2}\left(n+\kappa \right)-n$. The parameter α reflects the degree of dispersion of the sample points around the mean. The larger the value of α, and the further the sample points are from the mean value, the better the nonlinear characteristics are covered, but it may also introduce more unwanted noise. Here, α is taken as 1. The parameter κ=3−n.

The weights $w_{i}$ corresponding to the sampling points are as in Equations (19)–(21):(19)$$\begin{array}{cc} w_{0}^{\left(m\right)}=\frac{\lambda }{n+\lambda }, & i=0 \end{array}$$
(20)$$\begin{array}{cc} w_{0}^{\left(c\right)}=\frac{\lambda }{n+\lambda }+1-\alpha^{2}+\beta , & i=0 \end{array}$$
(21)$$\begin{array}{cc} w_{i}^{\left(m\right)}=w_{i}^{\left(c\right)}=\frac{1}{2\left(n+\lambda \right)}, & i=1,\ldots ,2n \end{array}$$
where the parameter β is used to approximate the prior distribution of X. Here, β=2. The superscript m denotes the weights in the state update and the superscript c denotes the weights in the covariance matrix update.

Step 3: Calculate the estimate of the state prediction ${\hat{X}}_{k,k-1}$ and the covariance matrix of the state prediction $P_{k,k-1}$:(22)$$\begin{array}{cc} \xi_{k,k-1}^{i}=f\left(\xi_{k-1}^{i}\right), & i=0,1,\cdots ,2n \end{array}$$
(23)$${\hat{X}}_{k,k-1}=\sum_{i=0}^{2n}w_{i}^{\left(m\right)}\xi_{k,k-1}^{i}$$
(24)$$P_{k,k-1}=\sum_{i=0}^{2n}w_{i}^{\left(c\right)}\left(\xi_{k,k-1}^{i}-{\hat{X}}_{k,k-1}\right)\cdot {\left(\xi_{k,k-1}^{i}-{\hat{X}}_{k,k-1}\right)}^{T}+Q_{k-1}$$

Step 4: Calculate the measurement prediction ${\hat{Z}}_{k,k-1}$ and the corresponding covariance matrix $P_{{\hat{Z}}_{k}}$, as well as the interaction covariance matrix $P_{{\hat{X}}_{k}{\hat{Z}}_{k}}$ of the measurement and state vectors.

According to the measurement equation, the predicted value of the measurement at point δ can be obtained as shown in Equation (25):(25)$$\varsigma_{k,k-1}^{i}=h\left(\xi_{k,k-1}^{i}\right)$$

Then the measurement prediction ${\hat{Z}}_{k,k-1}$ and the corresponding covariance matrix $P_{{\hat{Z}}_{k}}$ are as shown in Equations (26) and (27):(26)$${\hat{Z}}_{k,k-1}=\sum_{i=0}^{2n}w_{i}^{\left(m\right)}\varsigma_{k,k-1}^{i}$$
(27)$$P_{{\hat{Z}}_{k}}=\sum_{i=0}^{2n}w_{i}^{\left(c\right)}\left(\varsigma_{k,k-1}^{i}-{\hat{Z}}_{k,k-1}\right){\left(\varsigma_{k,k-1}^{i}-{\hat{Z}}_{k,k-1}\right)}^{T}+R_{k}$$

The interaction covariance matrix $P_{{\hat{X}}_{k}{\hat{Z}}_{k}}$ of the measurement and state vectors is:(28)$$P_{{\hat{X}}_{k}{\hat{Z}}_{k}}=\sum_{i=0}^{2n}w_{i}^{\left(c\right)}\left(\xi_{k,k-1}^{i}-{\hat{X}}_{k,k-1}\right){\left(\varsigma_{k,k-1}^{i}-{\hat{Z}}_{k,k-1}\right)}^{T}$$

Step 5: Calculate filter gain $K_{k}$:(29)$$K_{k}=P_{{\hat{X}}_{k}{\hat{Z}}_{k}}{P_{{\hat{Z}}_{k}}}^{-1}$$

Step 6: Calculate the state update equation and state update covariance matrix.

If the measurement obtained by the sensors at the k-th sampling instant is $Z_{k}$, then the state estimate and the state estimate covariance matrix at the k-th sampling instant are as follows:(30)$${\hat{X}}_{k}={\hat{X}}_{k,k-1}+K_{k}\left(Z_{k}-{\hat{Z}}_{k,k-1}\right)$$
(31)$$P_{k}=P_{k,k-1}-K_{k}P_{{\hat{Z}}_{k}}K_{k}^{T}$$

The above are the general steps of UKF. Due to the high degree of nonlinearity of the system in this paper, the symmetric sampling method is used to approximate the probability density distribution of the nonlinear function, which is used to approximate the posterior distribution of the nonlinear system.

## 4. Adaptive Unscented Kalman Filter 

With uncertainty in the statistical properties of the measurement noise, computer rounding errors can cause the state estimation covariance matrix $P_{k}$ and state prediction covariance matrix $P_{k,k-1}$ to lose their non-negative qualities and symmetry as the UKF calculation progresses, which can distort the filter gain matrix $K_{k}$ calculation and cause the filter to diverge. For this case, this paper introduces an adaptive noise factor to continuously adjust the measurement noise covariance matrix $R_{k}$ in order to reduce the impact of the uncertainty in the statistical characteristics of the measurement noise on the system.

Assuming that, after the introduction of the adaptive noise factor $\sigma_{k}$, the measurement prediction covariance matrix is ${\tilde{P}}_{{\hat{Z}}_{k}}$, then:(32)$${\tilde{P}}_{{\hat{Z}}_{k}}=\sum_{i=0}^{2n}w_{i}^{\left(c\right)}\left(\varsigma_{k,k-1}^{i}-{\hat{Z}}_{k,k-1}\right){\left(\varsigma_{k,k-1}^{i}-{\hat{Z}}_{k,k-1}\right)}^{T}+\sigma_{k}R_{k}$$

Let ${\tilde{Z}}_{k}$ be the difference between the measurement vector and its prediction, then:(33)$${\tilde{Z}}_{k}=Z_{k}-{\hat{Z}}_{k,k-1}$$

${\tilde{Z}}_{k}$ is the innovation vector, also known as the predicted residual vector. Based on Sage’s moving window estimation method, the innovation covariance matrix can be estimated as in Equation (34):(34)$${\hat{P}}_{{\tilde{Z}}_{k}}=\frac{1}{N}\sum_{i=0}^{N-1}{\tilde{Z}}_{k-i}{\tilde{Z}}_{k-i}^{T}$$
where N is the fixed window length.

The moving window estimation method provides information about the external noise based on the average of a specified number of innovation covariance matrices. The Adaptive Unscented Kalman Filter (AUKF) is based on this information to adjust the measurement noise covariance. Thus, the measurement prediction covariance matrix $R_{k}$ is corrected to achieve the effect of adaptive filtering.

The relationship between the innovation covariance matrix estimated from the mean of the innovation covariance matrices at the N sampling moments and the covariance matrix of the measurement prediction corrected by the adaptive factor $\sigma_{k}$ should satisfy Equation (35): (35)$${\tilde{P}}_{{\hat{Z}}_{k}}={\hat{P}}_{{\tilde{Z}}_{k}}$$

According to Equation (35), the relationship between the adaptive factor $\sigma_{k}$ and the innovation covariance matrix and the measurement prediction covariance matrix not corrected by the adaptive factor $\sigma_{k}$, as well as the measurement noise covariance matrix $R_{k}$, can be obtained as Equation (36):(36)$$\sigma_{k}R_{k}={\hat{P}}_{{\tilde{Z}}_{k}}-P_{{\hat{Z}}_{k}}+R_{k}$$

Simultaneously, compute the traces of both sides of Equation (36). The expression for the adaptive noise factor $\sigma_{k}$ can be obtained as in Equation (37):(37)$$\sigma_{k}=1-\frac{tr\left(P_{{\hat{Z}}_{k}}\right)-tr\left({\hat{P}}_{{\tilde{Z}}_{k}}\right)}{tr\left(R_{k}\right)}$$
where $tr\left(\cdot \right)$ is the trace computing symbol. In practical systems, an upper limit needs to be devised for the adaptive noise factor. If the adaptive noise factor is too large, the system will become too sensitive to changes in the real noise and will produce a large response to small changes in noise. If the upper limit of the adaptive noise factor is set too low, the system will be less able to adapt to the noise. However, the exact value of this upper limit is not universal and needs to be determined according to the actual application scenario and the performance requirements of the filter.

To effectively alleviate this problem, we can use historical data or experimental data to estimate the reasonable range of noise factors and set the upper limit accordingly. At the same time, the effect of the adaptive noise factor is verified through simulation or actual testing, including its stability and accuracy. The upper limit of the noise factor can be adjusted according to the test results to optimize the performance of the filter.

The gain of the Unscented Kalman filter after correction for the adaptive noise factor is given by Equation (38):(38)$${\bar{K}}_{k}=P_{{\hat{X}}_{k}{\hat{Z}}_{k}}{{\tilde{P}}_{{\hat{Z}}_{k}}}^{-1}$$

The state estimate ${\hat{X}}_{k}$ and the state estimate covariance matrix $P_{k}$ at the k-th sampling instant are given in Equations (39) and (40):(39)$${\hat{X}}_{k}={\hat{X}}_{k,k-1}+{\bar{K}}_{k}\left(Z_{k}-{\hat{Z}}_{k,k-1}\right)$$
(40)$$P_{k}=P_{k,k-1}-{\bar{K}}_{k}{\tilde{P}}_{{\hat{Z}}_{k}}{\bar{K}}_{k}^{T}$$

## 5. BCRLB Lower Bound Analysis

The Bayesian Cramér-Rao Lower Bound (BCRLB) is a performance criterion that can be used to describe the theoretical lower bound of the error covariance matrix of the Adaptive Unscented Kalman filter. The covariance of any unbiased Bayesian estimator cannot fall below this bound. It is given in Equation (41):(41)$$E\left[\left(X_{k}-{\hat{X}}_{k}\right){\left(X_{k}-{\hat{X}}_{k}\right)}^{T}\right]\geq J{\left(X_{k}\right)}^{-1}$$
where $E\left[\cdot \right]$ is the error covariance matrix of the Adaptive Unscented Kalman filter, and $J\left(X_{k}\right)$ is the spatial target state Fisher information matrix. $J\left(X_{k}\right)$ can be expressed as the sum of the a priori information and the data information of the spatial target, as shown in Equation (42):(42)$$J\left(X_{k}\right)=J_{P}\left(X_{k}\right)+J_{D}\left(X_{k}\right)$$
where $J_{P}\left(X_{k}\right)$ is the a priori information matrix and $J_{D}\left(X_{k}\right)$ is the data information matrix of the spatial target. Since the motion model of the spatial target is nonlinear, the calculation of the a priori information matrix $J_{P}\left(X_{k}\right)$ and the data information matrix $J_{D}\left(X_{k}\right)$ of the spatial target requires the linearization of the nonlinear equations of state and measurement (Equation (13)), as shown in Equation (43).
(43)$$\left\{\begin{array}{c} X_{k}=F_{k-1}X_{k-1}+V_{k-1}\\ Z_{k}=H_{k}X_{k}+W_{k} \end{array}\right.$$
where $F_{k}$ is the linearized state transfer matrix and $H_{k}$ is the linearized measurement matrix.
(44)$$F_{k}={\left.\frac{\partial f\left(X\right)}{\partial X}\right|}_{X=X_{k}}$$
(45)$$H_{k}={\left.\frac{\partial h\left(X\right)}{\partial X}\right|}_{X=X_{k}}$$

Then, the recursive formula for the a priori information matrix $J_{P}\left(X_{k}\right)$ is shown in Equation (46):(46)$$J_{P}\left(X_{k}\right)={\left(Q_{k-1}+F_{k-1}J^{-1}\left(X_{k-1}\right)F_{k-1}^{T}\right)}^{-1}$$

The data information matrix $J_{D}\left(X_{k}\right)$ is given by Equation (47). The expectation value needs to be calculated.
(47)$$J_{D}\left(X_{k}\right)=E\left(H_{k}^{T}R_{k}^{-1}H_{k}\right)$$

To simplify the calculations, the expectation value is usually approximated by the predicted values of the statistical averages, i.e., ${\hat{X}}_{k,k-1}$ is used instead of $X_{k}$, as shown in Equation (48):(48)$$J_{D}\left(X_{k}\right)={\left.H_{k}^{T}R_{k}^{-1}H_{k}\right|}_{X={\hat{X}}_{k,k-1}}$$

From the above, the Fisher information matrix $J\left(X_{k}\right)$ of the spatial target state can be derived, from which the BCRLB can be calculated, i.e., $J^{-1}\left(X_{k}\right)$. When the estimated covariance of the filter approaches the BCRLB, it can be inferred that the filter’s performance is nearing its optimal state. 

Then, Equation (49) can be adopted as a measure of spatial target tracking performance. The standard symbol is denoted as l:(49)$$l=tr\left(J^{-1}\left(X_{k}\right)\right)$$

While BCRLB offers a theoretical lower bound for assessing estimator performance, it does not directly dictate the value of the adaptive noise factor. The adaptive noise factor proposed in this article, whose value is influenced by measurement covariance, is used to adjust the measurement noise covariance matrix in real-time, thereby better accommodating the dynamic changes in the system.

In scenarios where the statistical characteristics of noise are unknown or time-varying, AUKF may exhibit superior accuracy and robustness due to its adaptive adjustment of the noise factor. Conversely, UKF may struggle to adapt promptly to noise variations, potentially leading to performance degradation. Regardless of whether UKF or AUKF, BCRLB serves as a theoretical benchmark for evaluating their performance. However, due to AUKF’s adaptive capabilities, it may more readily approach the performance threshold set by BCRLB in practical applications.

## 6. Experiments and Analyses

The tracking problem of a spatial target is simulated with a collaborative detection of the space-based infrared satellite and the ground-based radar using the Unscented Kalman Filter and the Adaptive Unscented Kalman Filter algorithm under the condition that both the process noise and the measurement noise are Gaussian white noise, while the process noise is Gaussian white noise with fixed mean and the measurement noise is time-varying noise.

It is assumed that the ground-based radar coordinates in the OXYZ coordinate system are $\left(3\times 10^{6}\;m,3\times 10^{6}\;m,1.5\times 10^{6}\;m\right)$. The initial state vectors of the space-based infrared satellite and the spatial targets are shown in Table 1. The total observation duration of the space-based infrared satellite and the ground-based radar is T=10000 s.

According to the spatial target motion model, the motion trajectories of the infrared satellite and the spatial targets are plotted as shown in Figure 1.

Set the initial state estimation error ΔX, as in Equation (50).
(50)$$\Delta X=[10^{3}\;10^{3}\;10^{3}\;1\;1\;1]$$

Set the initial covariance matrix $P_{0}$, as in Equation (51).
(51)$$P_{0}=diag\left(\left[10^{6}\;10^{6}\;10^{6}\;1\;1\;1\right]\right)$$

The process noise covariance matrix Q is assumed to be fixed, as in Equation (52).
(52)$$Q=diag\left(\left[10^{-6}\;10^{-6}\;10^{-6}\;10^{-10}\;10^{-10}\;10^{-10}\right]\right)$$

When the measurement noise is assumed to be Gaussian white noise with constant covariance, as in Equation (53):(53)$$R=diag\left(\left[0.09^{2}\;0.09^{2}\;1\times 10^{4}\;0.09^{2}\;0.09^{2}\right]\right)$$

When the measurement noise is assumed to be time-varying Gaussian white noise, as in Equation (54):(54)$$R=\left\{\begin{array}{c} \begin{array}{cc} diag([0.09^{2}\;0.09^{2}\;1\times 10^{4}\;0.09^{2}\;0.09^{2}]), & \;1\leq t\leq 2000\\ diag\left(\left[0.18^{2}\;0.18^{2}\;2\times 10^{4}\;0.3^{2}\;0.2^{2}\right]\right), & \;2001\leq t\leq 4000\\ diag\left(\left[0.2^{2}\;0.6^{2}\;3\times 10^{4}\;0.2^{2}\;0.09^{2}\right]\right), & \;4001\leq t\leq 6000\\ diag\left(\left[1^{2}\;0.4^{2}\;2\times 10^{4}\;0.5^{2}\;0.2^{2}\right]\right), & 6001\leq t\leq 8000\\ diag\left(\left[0.2^{2}\;1^{2}\;3\times 10^{4}\;1^{2}\;0.09^{2}\right]\right), & 8001\leq t\leq 10000 \end{array} \end{array}\right.$$

### 6.1. Gaussian White Noise with Constant Covariance

After 100 Monte Carlo trials, the root mean square error (RMSE) of tracking spatial target 1 with Gaussian white noise with constant covariance is shown in Figure 2, while the partial enlarged plot of the position root mean square error is shown in Figure 3.

The root mean square error (RMSE) of tracking spatial target 2 with Gaussian white noise with constant covariance is shown in Figure 4, while the partial enlarged plot of the position root mean square error is shown in Figure 5.

When the measurement noise is Gaussian white noise with constant covariance, the upper limit of the adaptive noise factor in the AUKF is set to 1, as in Equation (55):(55)$$\sigma_{k}\leq 1$$

Equation (55) applies to the case where the measurement noise is constant or with small variations.

### 6.2. Gaussian White Noise with a Step Change in Covariance

In the case where the measurement noise is Gaussian white noise with a step change in covariance, the tracking RMSE of the spatial target 1 is shown in Figure 6. The partial magnification of the position RMSE is shown in Figure 7.

In the case where the measurement noise is Gaussian white noise with a step change in covariance, the tracking RMSE of the spatial target 2 is shown in Figure 8. The partial magnification of the position RMSE is shown in Figure 9.

When the measurement noise is Gaussian white noise with step change in covariance, the upper limit of the adaptive noise factor in the AUKF is set to 1.5, as in Equation (56): (56)$$\sigma_{k}\leq 1.5$$

Equation (56) applies to situations where the measurement noise changes more than before. It takes into account changes in the measurement noise while avoiding oversensitivity to it.

### 6.3. Experimental Analysis

The RMSE at the k-th instant in the x-direction ($RMSE_{x_{k}}$) is calculated as in Equation (57): (57)$$RMSE_{x_{k}}=\sqrt{\frac{1}{M}\sum_{i=1}^{M}{\left(x_{k}-{\hat{x}}_{k}\right)}^{2}}$$
where M is the number of Monte Carlo simulations and $x_{k}$ is the position component of the state vector $X_{k}$ in the x-direction at the k-th instant.
(58)$$X_{k}={\left[x_{k}\;y_{k}\;z_{k}\;vx_{k}\;vy_{k}\;vz_{k}\right]}^{T}$$

Similarly, the RMSE at the k-th instant in the y-direction ($RMSE_{y_{k}}$) and the RMSE at the k-th instant in the z-direction ($RMSE_{z_{k}}$) can be calculated.

The RMSE of the position at the k-th instant ($RMSE_{r_{k}}$) is given by Equation (59):(59)$$\begin{array}{cc} RMSE_{r_{k}}= & \sqrt{RMSE_{x_{k}}^{2}+RMSE_{y_{k}}^{2}+RMSE_{z_{k}}^{2}}\\ = & \sqrt{\frac{1}{M}\sum_{i=1}^{M}\left[{\left(x_{k}-{\hat{x}}_{k}\right)}^{2}+{\left(y_{k}-{\hat{y}}_{k}\right)}^{2}+{\left(z_{k}-{\hat{z}}_{k}\right)}^{2}\right]} \end{array}$$

The average RMSE for all times in the x-direction is calculated as in Equation (60) and, similarly, the average RMSE can be calculated in the y-direction and z-direction, as well as in the overall position:(60)$$RMSE_{x}=\sqrt{\frac{1}{T}\sum_{j=1}^{T}{RMSE_{x_{j}}}^{2}}$$

The average RMSE of the two methods for different noise scenarios is calculated as shown in Table 2.

Combining Figure 2, Figure 3, Figure 4 and Figure 5 and Table 2, where the algorithms of Unscented Kalman Filter and Adaptive Unscented Kalman Filter are compared, it can be seen that AUKF reduces the tracking error of the spatial target 1 by about 1 m compared to UKF when the measurement noise covariance is constant. However, when the measurement noise remains constant, the average error of the UKF for space target 2 is marginally smaller than that of the AUKF. However, as the duration of the experiment increases, the error of the AUKF consistently remains smaller than that of the UKF.

Monte Carlo simulation experiments show that the improvement in the AUKF algorithm over the UKF algorithm is not very large when the observations are subjected to statistically stable perturbations. Overall, the AUKF algorithm has a slightly smaller target tracking error than the UKF algorithm in this condition.

Combining Figure 6, Figure 7, Figure 8 and Figure 9 and Table 2, the Monte Carlo simulation experiments show that the AUKF algorithm reduces the tracking error of the spatial target 1 by about 2 m compared to the UKF algorithm and reduces the tracking error of the spatial target 2 by about 2 m compared to the UKF algorithm, for a step change in the measurement noise covariance. It can be seen that the tracking performance of the AUKF algorithm is significantly better than that of the UKF algorithm when the target motion is subject to large disturbances.

In particular, when comparing the tracking error of the UKF algorithm in the two cases of measurement noise, it can be seen that the tracking error of the UKF algorithm in the case of fixed measurement noise covariance is significantly smaller than that in the case of step transformation of the measurement noise covariance. However, when comparing the tracking errors of the AUKF algorithm in the two cases of measurement noise, it can be found that the tracking errors of the AUKF algorithm are very close to each other. This shows that the AUKF is able to maintain good stability and robustness when tracking targets with large variations in observation noise characteristics.

From the above experimental analyses, it can be concluded that AUKF proposed in this paper, which adopts the adaptive noise factor method to dynamically adjust the observation noise covariance R, is able to effectively improve the accuracy of spatial target tracking. The AUKF is also able to maintain better stability and robustness in the situation of large variations in noise characteristics.

It should be noted that the adaptive noise factor proposed in this paper needs to adjust its upper limit according to the actual situation. Generally, the upper limit of the adaptive noise factor can be taken as 1 when the variation of measurement noise characteristics is small, and the upper limit of the adaptive noise factor can be taken as 1.5 when the variation of measurement noise characteristics is large, which needs to be determined according to the actual situation.

## 7. Conclusions

When the measurement noise characteristics vary greatly, the UKF algorithm reduces the tracking accuracy for spatial targets. This paper proposes an adaptive UKF algorithm based on the adaptive noise factor method. A space-based infrared satellite and a ground-based radar are used for cooperative observation, and the corresponding observation models are established. Through Monte Carlo simulation experiments for spatial target tracking, it is verified that the AUKF algorithm has improved spatial target tracking accuracy and stability compared to UKF and has better robustness. 

## Figures and Tables

**Figure 1 sensors-24-06094-f001:**
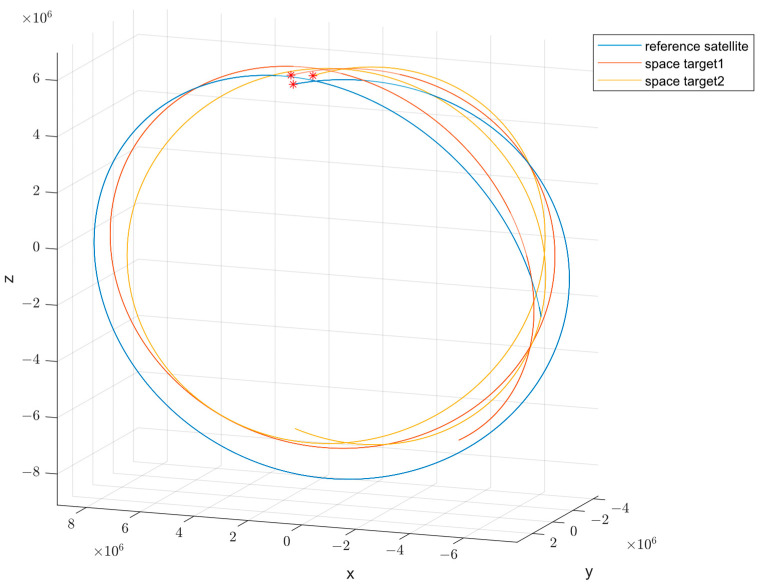
The Motion Trajectories of the Infrared Satellite and the Spatial Targets. The asterisks mean the initial point of the trajectory.

**Figure 2 sensors-24-06094-f002:**
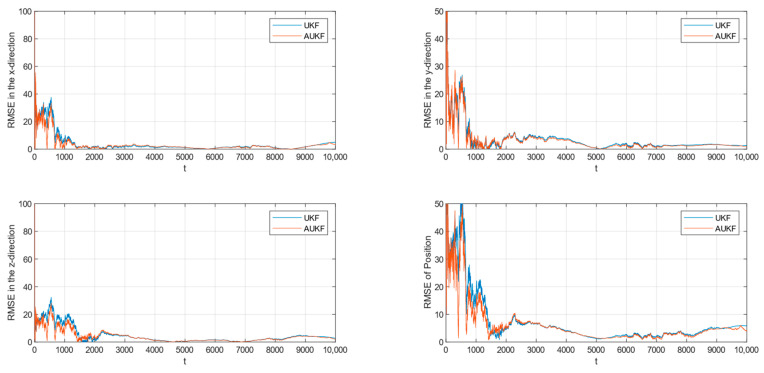
RMSE of tracking the spatial target 1 with fixed measurement noise covariance.

**Figure 3 sensors-24-06094-f003:**
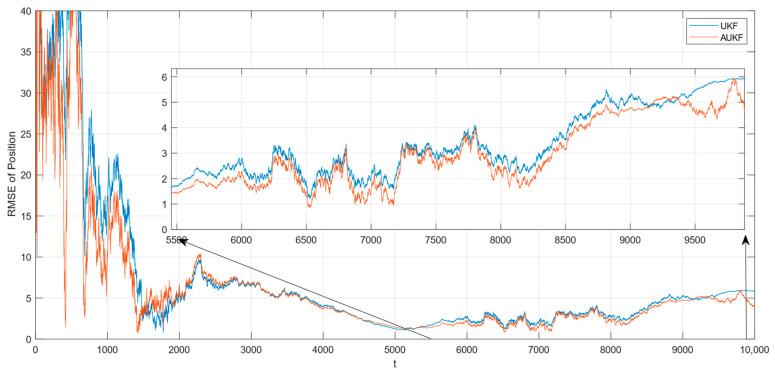
Amplification of the RMSE of position for tracking the spatial target 1 with a fixed covariance of the measurement noise.

**Figure 4 sensors-24-06094-f004:**
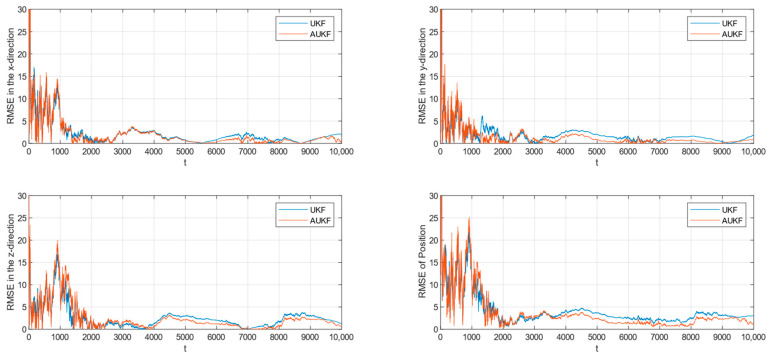
RMSE of tracking spatial target 2 with fixed measurement noise covariance.

**Figure 5 sensors-24-06094-f005:**
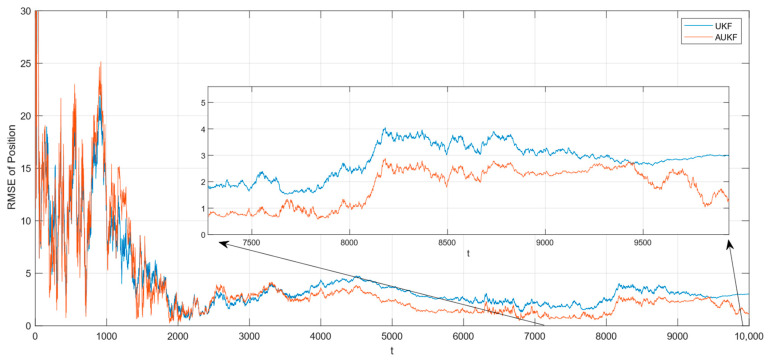
Amplification of the RMSE of position for tracking spatial target 2 with a fixed covariance of the measurement noise.

**Figure 6 sensors-24-06094-f006:**
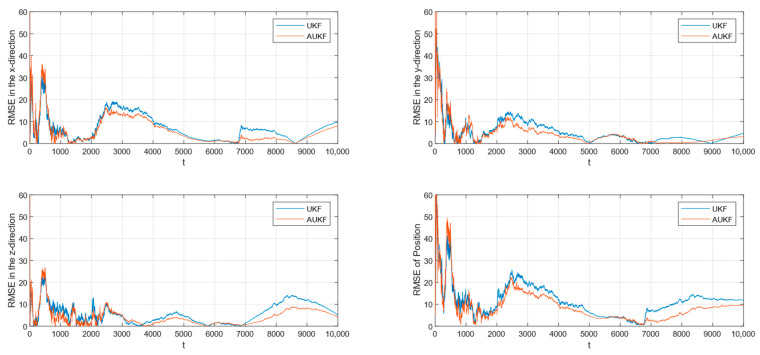
RMSE of tracking spatial target 1 with step change in measurement noise covariance.

**Figure 7 sensors-24-06094-f007:**
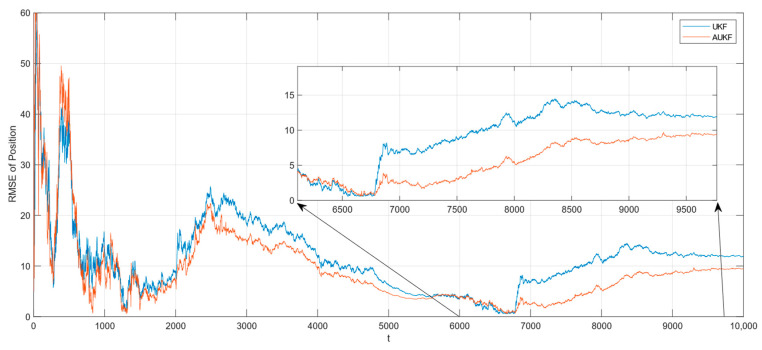
Zooming of position RMSE for tracking spatial target 1 with step change in measurement noise covariance.

**Figure 8 sensors-24-06094-f008:**
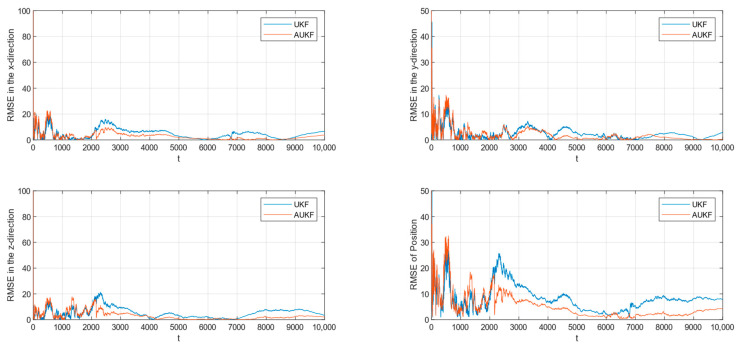
RMSE of tracking spatial target 2 with step change in measurement noise covariance.

**Figure 9 sensors-24-06094-f009:**
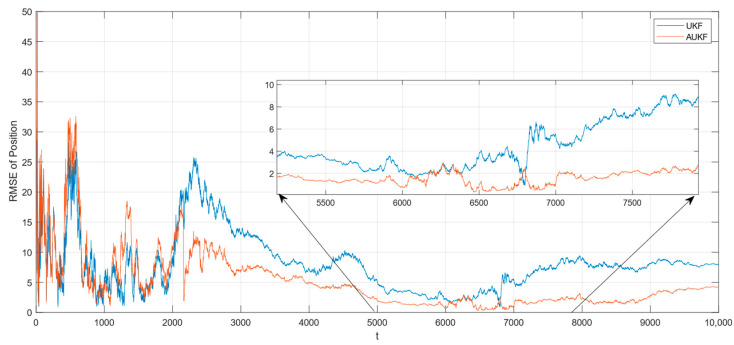
Zooming of position RMSE for tracking spatial target 2 with step change in measurement noise covariance.

**Table 1 sensors-24-06094-t001:** The initial state vectors of the space-based infrared satellite and the spatial target.

Name	x/m	y/m	z/m
The infrared satellite	$3.23\times 10^{5}$	$3.51\times 10^{6}$	$6.54\times 10^{6}$
The spatial target 1	$5.18\times 10^{5}$	$3.21\times 10^{6}$	$6.79\times 10^{6}$
The spatial target 2	$2.3\times 10^{-9}$	$2.42\times 10^{6}$	$6.65\times 10^{6}$
Name	$v_{x}/\left(m\cdot s^{-1}\right)$	$v_{y}/\left(m\cdot s^{-1}\right)$	$v_{z}/\left(m\cdot s^{-1}\right)$
The infrared satellite	$-6.78\times 10^{3}$	$-3.2\times 10^{3}$	$1.3\times 10^{3}$
The spatial target 1	$-6.35\times 10^{3}$	$-3.11\times 10^{3}$	$1.56\times 10^{3}$
The spatial target 2	$-5.76\times 10^{3}$	$-4.54\times 10^{3}$	$1.64\times 10^{3}$

The orbital elements corresponding to the satellite in Table 1 are as follows: Orbit Epoch: 20 August 2024 04:00:00.000 UTCG; Semimajor Axis: 8661.64 km; Eccentricity: 0.165076; Inclination: 66.007 deg; Argument of Perigee: 110.262 deg; RAAN: 57.3241 deg; Mena Anomaly: 334.318 deg.

**Table 2 sensors-24-06094-t002:** Comparison of the RMSE of the two algorithms (Unit: m).

The Spatial Target 1		Constant Measurement Noise Covariance		Step Change in Measurement Noise Covariance	
		UKF	AUKF	UKF	AUKF
Average RMSE	x	7.43	6.28	9.06	8.24
	y	5.92	5.97	7.47	6.65
	z	7.24	5.79	7.24	5.33
	r	11.95	10.43	13.80	11.79
**The Spatial Target 2**		**Constant Measurement Noise Covariance**		**Step Change in Measurement Noise Covariance**	
		**UKF**	**AUKF**	**UKF**	**AUKF**
Average RMSE	x	3.47	3.53	6.02	4.53
	y	3.19	3.35	3.52	3.29
	z	3.70	3.92	6.57	4.61
	r	5.99	6.25	9.59	7.26

## Data Availability

The data are contained within this article.

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
