# Peer review of "An Adaptive Spatial Target Tracking Method Based on Unscented Kalman Filter"

_sensors, 2024, doi:10.3390/s24186094_

Round 1
Reviewer 1 Report
Comments and Suggestions for Authors
This paper proposes an adaptive noise factor method based on Unscented Kalman filter to adaptively adjust the covariance matrix of the measurement noise, reducing the influence of time-varying measurement noise on the spatial target tracking. The results of numerical simulations proves the advantage of the proposed method compared to traditional UKF. It has certain academic and engineering value, but there are some things that might be better with further refinement and consideration:
1. The images in this article can be further optimized. The position of Figure 1 should be centered and adjusted to a more appropriate aspect ratio. All images will not be clear when enlarged. It is recommended to use vector images.
2. In section 5, the Bayesian Cramér-Rao Lower Bound (BCRLB) is introduced. What is its relationship with the adaptive noise factor proposed in this paper? Does it affect the accuracy of AUKF compared to UKF?
3. The upper bound of the adaptive factor is selected manually based on the empirical value. It would be better to explain the rationale behind this, or to give a more specific and quantitative approach to this upper bound adjustment.
Reviewer 2 Report
Comments and Suggestions for Authors
Given the high nonlinearity of the space target motion model and the time-varying measurement noise, this paper proposes an adaptive noise factor method based on the unscented Kalman filter to adaptively adjust the covariance matrix of the measurement noise. Simulation experiments show that this method reduces the impact of time-varying measurement noise on space target tracking, and can accurately represent the posterior mean and covariance of the space target state vector. The process is innovative and insightful.
Comments on the Quality of English Language1. On page 3, line 118, I think there should be a comma after the symbol 'V'.
2. On page 4, line 164, 'Firty' should be 'Firstly'.
Reviewer 3 Report
Comments and Suggestions for Authors
This study proposes a method to improve the accuracy of space target tracking based on the Unscented Kalman Filter (UKF) by adapting to time-varying measurement noise. The proposed method introduces an adaptive noise factor to adjust the covariance matrix of the measurement noise dynamically. Numerical simulations using ground-based radar and space-based infrared satellite measurement models demonstrate that the proposed method enhances space target tracking accuracy compared to the standard UKF. Although the proposed method slightly improved the accuracy, the numerical simulation is examined for one test case. I would like to know the simulation results for other orbit and have the following comments and questions.
- Equation (7) indicates the Runge-Kutta method. However, it would be not essential for the proposed method. In other words, other integration methods can be applicable to predict the state. Thus I think this formulation is not necessary.
- This paper focuses on measurement noise only. However, process noise is also difficult to set proper values. Can the adaptive noise factor method be applied to process noise as well?
- The satellite position is written with ECEF coordinates. But this is not intuitive. Orbital elements corresponding (x,y,z) coordinates should be written for better understanding.
- The number of Monte Carlo trials is 50 and seems small. How did you decide the number of Monte Carlo runs? Could this affect the reliability of the results? I think more Monte Carlo trials are required.
Round 2
Reviewer 3 Report
Comments and Suggestions for Authors
Properly revised.